# Host-targeted niclosamide inhibits *C. difficile* virulence and prevents disease in mice without disrupting the gut microbiota

John Tam[1,2], Therwa Hamza[3], Bing Ma[4], Kevin Chen [3], Greg L. Beilhartz[1], Jacques Ravel [4], Hanping Feng[3] & Roman A. Melnyk[1,2]

*Clostridium difficile* is the leading cause of nosocomial diarrhea and colitis in the industrialized world. Disruption of the protective gut microbiota by antibiotics enables colonization by multidrug-resistant *C. difficile*, which secrete up to three different protein toxins that are responsible for the gastrointestinal sequelae. Oral agents that inhibit the damage induced by toxins, without altering the gut microbiota, are urgently needed to prevent primary disease and break the cycle of antibiotic-induced disease recurrence. Here, we show that the anthelmintic drug, niclosamide, inhibits the pathogenesis of all three toxins by targeting a host process required for entry into colonocytes by each toxin. In mice infected with an epidemic strain of *C. difficile*, expressing all three toxins, niclosamide reduced both primary disease and recurrence, without disrupting the diversity or composition of the gut microbiota. Given its excellent safety profile, niclosamide may address an important unmet need in preventing *C. difficile* primary and recurrent diseases.

[1] Molecular Medicine, Hospital for Sick Children, 686 Bay St., Toronto, ON M5G 0A4, Canada. [2] Department of Biochemistry, University of Toronto, Toronto, ON M5S 1A8, Canada. [3] Department of Microbial Pathogenesis, University of Maryland Dental School, Baltimore, MD 21201, USA. [4] Institute for Genome Sciences, University of Maryland School of Medicine, Baltimore, MD, USA. Correspondence and requests for materials should be addressed to R.A.M. (email: roman.melnyk@sickkids.ca)

**C**lostridium difficile is a spore-forming Gram-positive bacterium that causes a range of gastrointestinal diseases, typically in individuals that have taken a course of broad-spectrum antibiotics, which lowers the diversity of the protective resident GI microbiota[1]. In the resulting aftermath, opportunistic C. difficile colonizes the lower GI tract of susceptible individuals and secretes up to three gut-damaging toxins, including two large homologous toxins TcdA and TcdB, and in the case of epidemic strains of C. difficile—such as ribotype 027 (RT027)[2] and ribotype 078 (RT078)[3]—a third, unrelated binary toxin called CDT[4]. In recent years, C. difficile has become a major public health concern, due to the proliferation and global spread of epidemic strains, which are associated with increased morbidity and mortality[5]. The increased virulence of these strains has been attributed to several factors, including acquisition of mutations in gyrB that result in resistance to fluoroquinolones[5]; increased expression of toxins[6,7]; production of a more cytotoxic form of TcdB[8]; and, expression of CDT[2].

Extensive experimental and epidemiological evidence support a role for toxins as the primary determinants of disease pathogenesis. Isogenic knockout studies, in which toxins were deleted individually, or in combination, showed that TcdA and TcdB alone are sufficient to cause fulminant disease in hamsters[9,10], whereas CDT appears to contribute to virulence in combination with TcdA or TcdB[11]. Importantly, knockout of all three toxins renders C. difficile completely avirulent.

The inextricable link between antibiotic-induced dysbiosis of the GI microbiota and infection by C. difficile, together with the well-validated role of toxins in driving disease pathogenesis, provide strong rationale and validation for targeting the actions of the C. difficile toxins as a novel approach to treat or prevent C. difficile infection (CDI). Despite early clinical setbacks with nonspecific polymers meant to sequester the toxins in the GI tract[12], the monoclonal antibody bezlotoxumab (Zinplava—Merck), which binds to and blocks TcdB following toxin-induced damage of the gut lining[13], was recently approved for use in CDI patients for reducing recurrence[14,15]. This important clinical validation of toxin-targeting approaches for treating C. difficile recurrence has fueled efforts to develop next-generation anti-toxins that are orally-bioavailable (i.e., small molecules), have a greater spectrum of activity against all C. difficile toxins, and potentially be used prior to, or during a suspected primary infection. Ideally, such a therapeutic, in addition to having an impeccable safety profile in humans, would not itself affect the composition of the protective gut microbiota, which is ultimately required to prevent further re-infection. Moreover, with the emergence of new C. difficile ribotypes, such as RT033, that do not express TcdB, but are nevertheless pathogenic[16–20], it would be desirable to have a single agent with the above characteristics that is also able to prevent TcdA- and/or CDT-induced pathogenesis.

In this study, we screened libraries of approved drugs in a phenotypic screen of TcdB-induced cell rounding with the goal of identifying small molecules that may potentially be repurposed for treating CDI through direct toxin inhibition. Recent drug repurposing phenotypic screens have led to discoveries of potential new candidate therapies for a number of infectious diseases, including for giardiasis[21], Zika virus infection[22], Ebola virus disease[23], and Hepatitis C infection[24]. Here, among the panel of hits identified that completely protected cells from TcdB intoxication, we identified niclosamide, a widely used anthelmintic drug approved by the US FDA for treating intestinal infections of tapeworms[25]. Based on its excellent safety profile[26,27], and its preferential biodistribution in the colon resulting from its poor absorption in the GI tract[26], we investigated niclosamide as an oral toxin-neutralizing treatment for CDI.

## Results

**High-throughput screen for inhibitors of TcdB intoxication.** Intoxication by TcdB toxin is a multistep process, involving four functionally-distinct toxin domains, and several host-factors and processes (Fig. 1a). The intoxication of cells leads first to cytopathic effects (i.e., rounding of cells)[28] within 1–3 h, and later, cytotoxic effects (i.e., apoptosis) after 24 h[29]. To identify small molecules that protected cells from TcdB, we employed a high-throughput assay of TcdB-induced cell rounding that we previously developed, which quantifies the extent of rounding of human lung fibroblasts treated with cytopathic doses of TcdB (i.e., 1 pM for 3 h) using high-content imaging analysis[30]. To increase the probability of identifying compounds with suitable properties for subsequent in vivo studies and beyond, we screened the Library of Pharmacologically Active Compounds (1280 compounds) and the Microsource Library Spectrum Collection (2360 compounds)—libraries consisting of approved drugs and pharmacologically active molecules with known targets and pharmacological properties.

From the 60 compounds that protected cells from TcdB by greater than the statistical cut-off of three standard deviations of the mean of the data (Supplementary Figure 1), we triaged drugs with undesirable mechanisms-of-action (i.e., antibiotics and antiseptics), and those that are known to be toxic or poorly tolerated in humans. Emerging from this prioritization were the three related salicylanilide anthelmintic drugs: niclosamide (71% inhibition), closantel (60% inhibition), and oxyclozanide (43% inhibition) (Fig. 1b, c); drugs that act on parasites within the GI lumen, and that have well-documented safety margins in humans. Among these salicylanilides, niclosamide was the most potent inhibitor of TcdB-induced cell-rounding, protecting cells with an $EC_{50} = 0.51 \pm 0.03\,\mu M$ (values represent mean ± s.e.m., $n = 5$) (Fig. 1d). Protection from TcdB-induced cell-rounding by niclosamide was complete; human IMR-90 fibroblasts that were co-incubated with niclosamide and TcdB were indistinguishable from cells that had not received toxin (Fig. 1e). To evaluate the extent of protection by niclosamide against different amounts of TcdB (reflecting the range of toxin levels that might be experienced during an infection), cells were treated with a range of TcdB concentrations at different fixed doses of niclosamide. In the absence of drug, TcdB dose-dependently induces cell rounding with an $EC_{50} = 0.8\,pM$ (Fig. 1f). In the presence of increasing concentrations of niclosamide, the amount of TcdB required to reach equivalent levels of rounding increased dramatically. Remarkably, above the $EC_{50}$ of niclosamide, cells were completely protected from TcdB by over three orders-of-magnitude, corresponding to a protection factor (PF) >5000 (Fig. 1f).

Next, we tested the ability of niclosamide and the more water-soluble ethanolamine salt form of niclosamide, niclosamide ethanolamine (NEN)[26] to maintain the integrity of human epithelial colorectal cells (CaCo-2 cells) that were treated with TcdB. Treatment of a confluent monolayer of CaCo-2 cells with TcdB results in disruption of monolayer integrity and loss of trans-epithelial resistance[31] within hours of application as a result of GTD-induced actin depolymerization. Disruption of the monolayer integrity by TcdB was prevented by co-treatment with niclosamide (Supplementary Figure 2). Furthermore, NEN prevented the TcdB-induced disruption of Caco-2 monolayers maintaining barrier function to untreated levels (Fig. 1g).

**Mechanism of TcdB neutralization by niclosamide.** To elucidate the mechanism by which niclosamide inhibits TcdB-induced cell rounding, we carried out a series of assays that evaluate each step of the intoxication pathway in isolation (Fig. 1a). Though

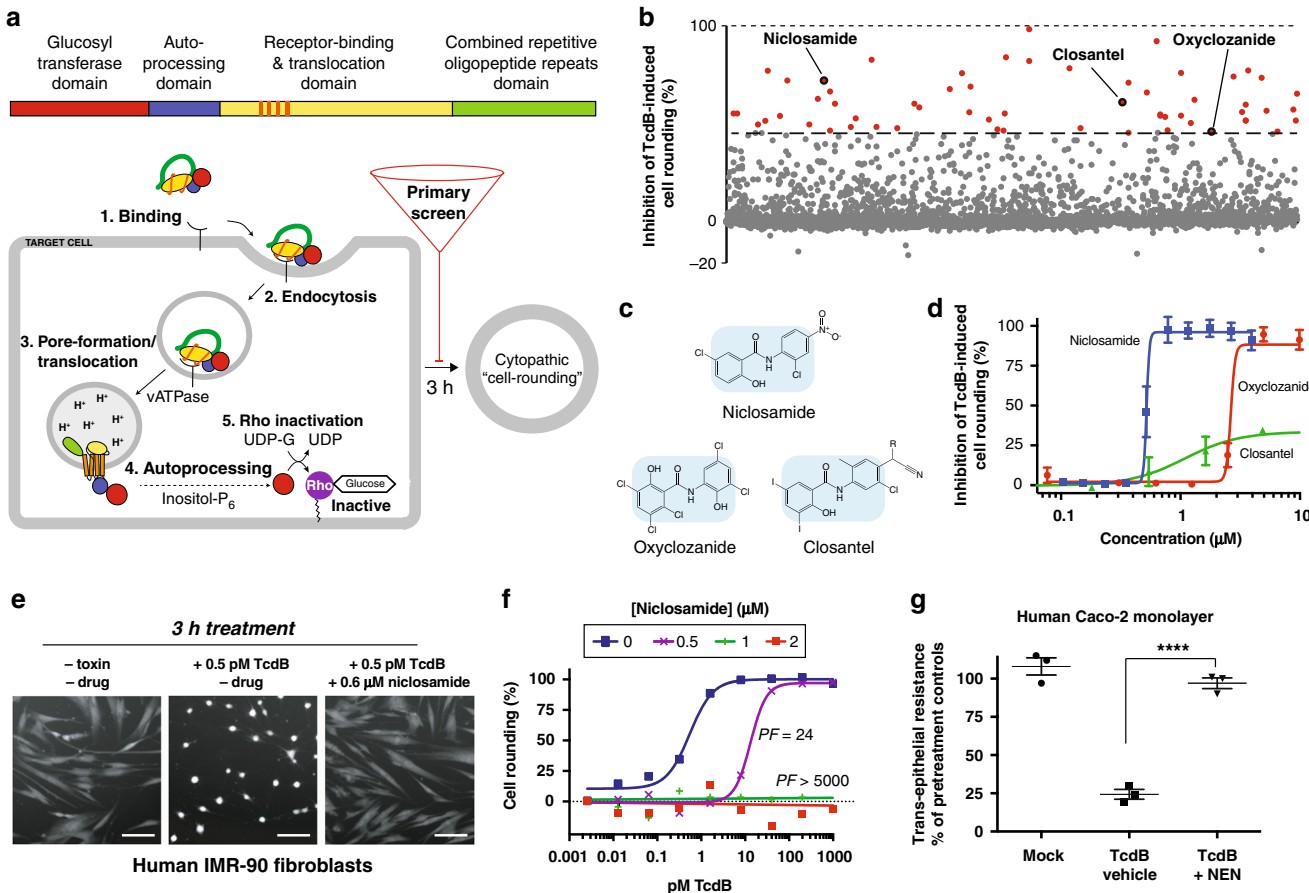

**Fig. 1** Identification of niclosamide from a high-throughput screen of TcdB-induced cell-rounding. **a** Top, domain architecture of TcdB, highlighting the four domains involved in TcdB-induced cell-rounding. Membrane-insertion elements that are thought to form the translocation pore within endosomes[33] are depicted as orange vertical lines within the receptor-binding and translocation domain. Bottom, steps involved in TcdB intoxication of human cells. Chemical libraries were screened at 40 μM using an algorithm developed previously[30]. **b** Results from high-throughput screening of 3580 drugs from the LOPAC and Microsource collections. A statistical cutoff of 43% inhibition of cell rounding was based on identification of molecules that were greater than 3-standard deviations above the mean of the data. The salicylanilide anthelmintic drugs niclosamide, closantel, and oxyclozanide were among the 60 hits identified. **c** Chemical structures of niclosamide, oxyclozanide, and closantel. **d** Dose titration curves of the inhibition of cell-rounding by niclosamide, oxyclozanide, and closantel ($n = 5$; $n = 3$ for clostantel). Values represent mean ± s.e.m. **e** Representative images of human IMR-90 fibroblasts from at least 8 experiments. Cells were pre-treated with DMSO or niclosamide 15 min before treatment with buffer or 0.5 pM TcdB and images were collected 3 h later. White bars represent 100 μm. **f** Intoxication of human IMR-90 cells by TcdB in the presence of different doses of niclosamide after 3-h. Protection factor, PF, represents the extent to which niclosamide shifts the curve for TcdB (i.e., $EC50_{niclosamide}/EC50_{vehicle}$). **g** Normalized trans-epithelial resistance measurements in human CaCo-2 cells, 6 h post-treatment. 5 μM NEN preserved significantly increased resistance across Caco-2 monolayer cells compared to mock control values ($n = 3$). Values represent mean ± s.e.m. ****$P < 0.0001$ (unpaired $t$-test with Welch's correction)

niclosamide dose-dependently inhibits Rac1 glucosylation in cells (Fig. 2a), no direct inhibition of GTD enzymatic activity is observed in vitro up to 10 μM niclosamide, indicating that inhibition occurs at a step upstream of Rac1 glucosylation (Fig. 2b). The release of GTD by the APD domain, the step immediately preceding Rac1 glucosylation, was also unaffected by up to 10 μM niclosamide in an in vitro assay of inositol hexakisphosphate-induced autoprocessing (Fig. 2c).

Consistent with niclosamide not acting through inhibition of either GTD or APD activity, we found that niclosamide inhibited TcdB-induced necrosis (Fig. 2d)—a cellular phenotype that is independent of GTD and APD activity, but dependent on cell-surface binding, uptake into endosomes, and pH-dependent pore-formation in endosomes for full activity[32,33]. Using mammalian cells expressing all three known receptors, we found that niclosamide did not affect TcdB binding to the cell surface (Fig. 2e). Moreover, we saw no evidence for a direct binding interaction between niclosamide and TcdB (Supplementary

Figure 3). Taken together, these data point to niclosamide acting on the host, and through inhibition of the pore-formation process. Niclosamide has been reported to mildly increase the pH of endosomes through a unique "proton-shuttle" mechanism[34], which is distinct from other modes of endosomal deacidification, such as lysosomotropism. Using fluorescent LysoTracker dye, a lysosomotropic molecule that accumulates in acidic vesicles, we found that pre-treating cells with either niclosamide or NEN indeed reduces LysoTracker fluorescent staining, consistent with these molecules increasing the pH of endosomal compartments (Fig. 2f). Importantly, the dose-dependent decrease in endosomal pH by niclosamide and NEN overlapped with the dose–titration curves for inhibition of cell rounding and necrosis (Fig. 2g).

**Niclosamide protects cells from all three *C. difficile* toxins.** The determination that niclosamide inhibited TcdB at the level of the host endosome prompted us to consider the intriguing possibility

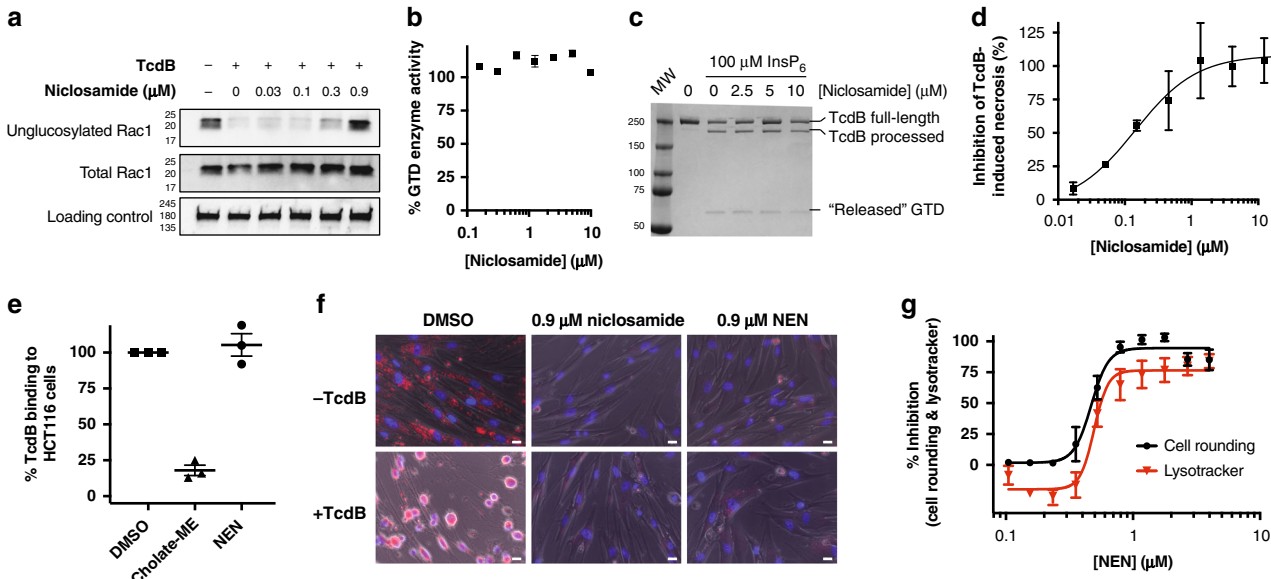

**Fig. 2** Mechanism-of-action of niclosamide inhibition of TcdB intoxication. **a** Western blot image for intracellular Rac1 glucosylation ($n = 3$). IMR-90 cells were treated with niclosamide (doses indicated) for 15 min, followed by treatment with 0.5 pM TcdB. Cells were harvested as described in Methods. Mab102, which recognizes un-glucosylated Rac1 in cells, shows a dose-dependent re-appearance with increasing concentration of niclosamide, relative to total Rac1 levels ($n = 2$). **b** In vitro GTD glucosyltransferase assay. Recombinant GTD was incubated for 30 min with different doses of niclosamide ($n = 3$). Activity was measured as described in Methods. Niclosamide at concentrations up to 10 μM did not directly affect GTD activity itself. **c** In vitro auto-processing assay. Compound or DMSO was added to 100 nM TcdB, and pre-incubated for 60 min. To initiate auto-processing, InsP6 was added (100 μM) and incubated @ 37 °C for 20 min before stopping with Laemlii sample buffer. Niclosamide at concentrations up to 10 μM did not affect auto-processing ($n = 2$). **d** Niclosamide was titrated in IMR90 cells in the presence of 1 nM TcdB for 3-h. Loss of cellular ATP as a marker of high dose toxicity was measured by adding CellTiter-Glo reagent (Promega). Niclosamide inhibited TcdB-induced necrosis with an $IC_{50} = 0.15$ μM. Bars represent SEM of two experiments. **e** Cell surface binding assay was carried out as described in Methods. Methylcholate, but not NEN, prevented surface binding of TcdB to cells ($n = 3$). **f** LysoTracker assay was carried out as described in Methods on IMR90 cells in the absence (top panels) or presence (bottom panels) of 1 pM TcdB. Acidic compartments are stained in red, while the nuclei are stained in blue. Cells treated with TcdB and vehicle are rounded and raised in comparison to untreated cells. Cells treated with niclosamide or NEN did not exhibit LysoTracker staining, and were protected from TcdB-induced cell rounding. White bars represent 20 μm. **g** Titration of NEN by LysoTracker Red DND-99. The LysoTracker fluorescence at ex/em 574/594 was measured on a Spectramax i3x plate reader (Molecular Devices). The $IC_{50}$s by cell rounding and LysoTracker assays were 0.47 and 0.49 μM, respectively. Bars represent SEM of four experiments

**Table 1 Inhibition of *C. difficile* toxin-induced cytotoxicity by niclosamide and NEN**

|  | TcdB$_{012}$ | TcdB$_{078}$ | TcdB$_{027}$ | TcdA | CDT |
|---|---|---|---|---|---|
| Niclosamide IC$_{50}$ (μM) | 0.44 ± 0.05 | 0.5 ± 0.2 | 0.4 ± 0.1 | 0.5 ± 0.1 | 0.53 ± 0.08 |
| NEN IC$_{50}$ (μM) | 0.43 ± 0.03 | 0.45 ± 0.09 | 0.5 ± 0.1 | 0.55 ± 0.08 | 0.50 ± 0.09 |

The data are expressed as the means ± s.e.m.
Niclosamide replicates: TcdB012, $n = 8$; TcdB078, $n = 4$; TcdB027, $n = 4$; TcdA, $n = 6$; CDT, $n = 4$
NEN replicates: TcdB012, $n = 7$; TcdB078, $n = 4$; TcdB027, $n = 3$; TcdA, $n = 4$; CDT, $n = 3$

that niclosamide might additionally block the actions of TcdA and CDT, both of which require endosomal acidification for pore-formation and intracellular entry. Indeed, niclosamide and NEN completely protected cells from TcdA-induced cell rounding, and from CDT-induced damage (i.e., depolymerization of the actin cytoskeleton), at the same doses that protect cells from TcdB (Table 1 and Supplementary Figure 4). NEN also protected cells against a form of TcdB derived from hypervirulent strains of *C. difficile*, which have been shown to enter cells at an earlier stage in endocytosis[8] (Table 1).

Demonstrating protection against CDT, a toxin that bears no structural or functional similarities to TcdA and TcdB, other than requiring low pH to escape endosomes, further supports the mechanism-of-action for niclosamide. More importantly, this finding suggests that niclosamide, as a single entity, could potentially protect from infection and disease by all pathogenic *C. difficile* strains, expressing any combination of toxins, in vivo.

**NEN reduces disease by an epidemic strain of *C. difficile* in mice.** Mice preconditioned with antibiotics and challenged with *C. difficile* develop typical CDI (weight loss, diarrhea, death) in the absence of any therapeutic countermeasures[1]. To test the hypothesis that niclosamide is capable of preventing disease induced by strains expressing multiple toxins, we evaluated the efficacy of NEN in protecting against CDI in a murine model challenged with the hypervirulent strain UK1 (RT027), which expresses TcdA, TcdB, and CDT. Infected mice were treated with either water (control) or NEN (at different doses; 2, 10, and 50 mg per kg) via oral gavage 4 h post spore challenge and for 3 consecutive days after spore challenge (Fig. 3a). Typical symptoms of CDI in murine models include severe weight loss on days 2 and 3 post-challenge accompanied with diarrhea and high mortality rate in sham groups. All doses of NEN tested significantly protected mice from weight loss compared to control group (Fig. 3b). NEN protected mice from death in a dose-dependent manner,

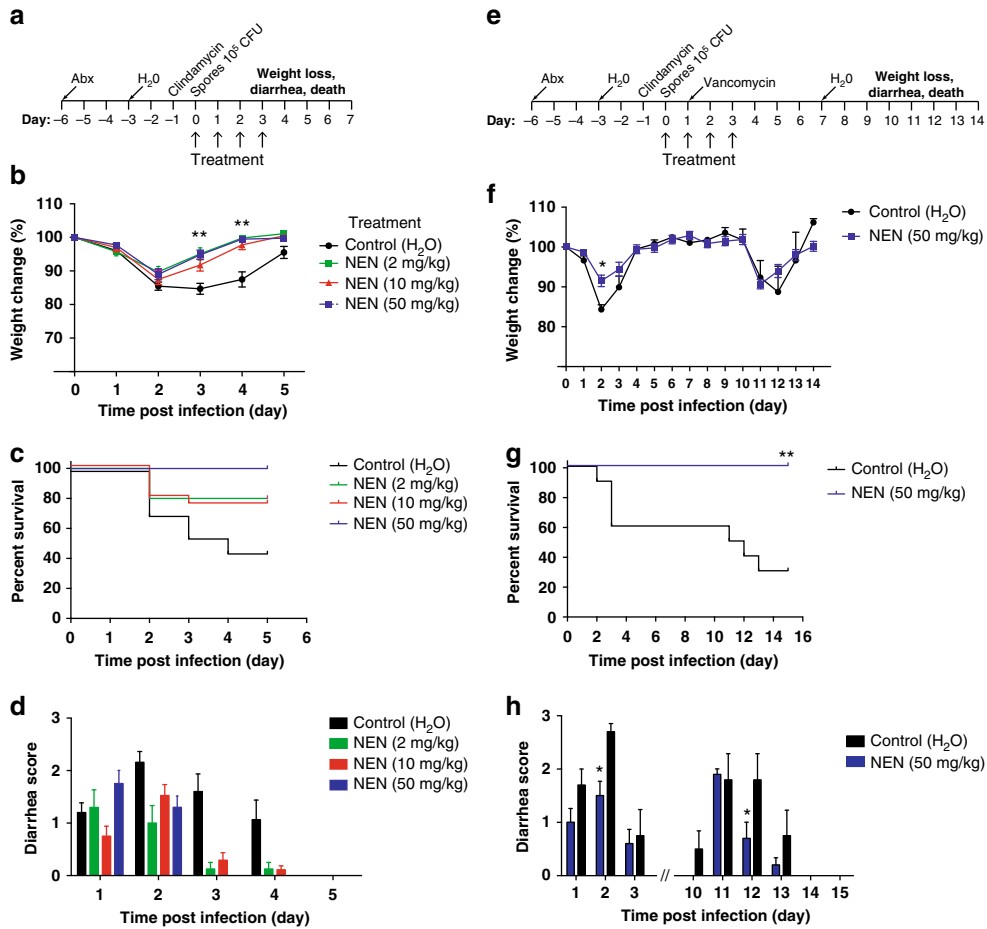

**Fig. 3** Niclosamide ethanolamine (NEN) is protective in primary and recurrent CDI. **a** Protocol schematic for primary CDI model. **b** Weights of mice after challenge with *C. difficile* spores ($10^5$ CFU/mL) on day 0. Mice were treated with vehicle (5% DMSO) or NEN (in 5% DMSO) at different doses (2, 10, and 50 mg per kg). Each point is the mean SE from day 0. **c** Mouse survival as determined by a log-rank/Mantel–Cox test. **d** Diarrhea score of infected mice[5]. **e** Protocol schematic for recurrent CDI model. **f** Weights of mice after *C. difficile* spore challenge (day 0) followed with vancomycin in their drinking water for recurrence CDI model. **g** Survival of infected mice treated and un-treated with NEN (50 mg per kg). **h** Diarrhea score of infected mice. *$p < 0.05$; **$p < 0.01$ using unpaired Student *t*-test. Data represent 10 mice/group

**Table 2 MIC values for niclosamide, NEN, and vancomycin on Clostridial species**

| Organism (strain) | Vancomycin (μg/mL) | NEN[a] (μg/mL) |
|---|---|---|
| *C. difficile* R20291 (027) | 1 | >19 |
| *C. difficile* M68 (017) | 2 | >19 |
| *C. clostridioforme* ATCC25537 | 4 | >19 |
| *C. sporogenes* ATCC3584 | 8 | >19 |

Data are $n = 3$
[a]19 μg/mL = 50 μM

with all mice in the 50 mg per kg group remarkably surviving infection, compared to only 45% for control group (Fig. 3c). These results closely tracked the wet tail and diarrhea scores, which were significantly lower in NEN-treated groups (Fig. 3d).

Symptomatic recurrence of CDI, which occurs in approximately one-in-four individuals, is a characteristic feature of CDI that complicates eradication and management of *C. difficile*[2]. We assessed whether NEN (50 mg per kg) could prevent recurrence in a mouse model of recurrent CDI in which infected mice are treated with vancomycin (0.5 mg per mL) in their drinking water starting on day 1 ongoing for 6 days after spore challenge (Fig. 3e). NEN treatment was given via oral gavage 4 h post spore challenge and for 3 consecutive days after spore challenge. Both

groups (NEN-treated and un-treated) started losing weight on day 4 after receiving vancomycin water (i.e., day 11 post spore challenge) (Fig. 3f). As above, all mice in the NEN group survived from *C. difficile* challenge, whereas more than 60% of mice in the control group became moribund (Fig. 3g, h). After resolution of symptoms, both groups began to lose weight at day 11; however, NEN-treated mice displayed less severe diarrhea scores, and importantly all NEN-treated mice survived recurrence.

**NEN does not affect *C. difficile* growth in vitro.** As salicylanilide derivatives have been reported previously to have antimicrobial activity against certain Gram-positive bacteria[35,36], we next carried out a series of experiments to address, whether NEN, specifically, had any antibacterial activity against *C. difficile* that may have contributed to the protective effects seen in vivo. To this end, we measured the minimum inhibitory concentrations (MICs) of NEN on individual strains of *Clostridium* species using the gold-standard anaerobic agar dilution assay[37]. No antimicrobial activity was seen for NEN up to 19 μg/mL (i.e., 50 μM NEN) against either of the two strains of *C. difficile* tested (017 and 027), or against the two non-pathogenic *Clostridium* species tested (Table 2). These results were confirmed against a larger panel of *C. difficile* strains, where we saw no activity for NEN up to 32 μg/mL (i.e., 84 μM) (Table 3). While these data indicate that

**Table 3 *C. difficile* MIC values for NEN, vancomycin, metronidazole, and fidaxomicin**

| Organism ATCC (MMX No.)[a] | MIC in µg/mL | | | |
|---|---|---|---|---|
| | Fidaxomicin | Vancomycin | Metronidazole | NEN |
| *C. difficile* 70005 (4381) | 0.06[b] | 4[c] | 0.5[d] | >32 |
| *C. difficile* (8261) | 0.06 | 2 | 0.5 | >32 |
| *C. difficile* (8262) | 0.12 | 4 | 0.25 | >32 |
| *C. difficile* (8263) | 0.06 | 2 | 0.25 | >32 |
| *C. difficile* (8341) | 0.06 | 1 | 0.25 | >32 |
| *C. difficile* (8336) | 0.12 | 4 | 0.5 | >32 |
| *C. difficile* (8337) | 0.06 | 1 | 0.25 | >32 |
| *C. difficile* (8338) | 0.12 | 4 | 2 | >32 |
| *C. difficile* (8339) | 0.12 | 4 | 4 | >32 |
| *C. difficile* (8340) | 0.12 | 2 | 0.5 | >32 |

[a]Micromyx Isolate Number
[b]CLSI QC range (0.06–0.25)
[c]CLSI QC range (0.5–4)
[d]CLSI QC range (0.125–0.5)

NEN does not affect *C. difficile* growth directly, an important feature to demonstrate for NEN, or any would-be *C. difficile* therapeutic, is the lack of effect on the gut microbiota.

**NEN effects on the gut microbiota in vivo**. To directly address whether NEN had any effects on the gut microbiota that may contribute to disease pathogenesis, we next evaluated the effect of NEN treatment on the composition and structure of the gut microbiota in mice under various situations. First, in mice that were not infected with *C. difficile*, we investigated the effects of NEN at the highest dose tested in the efficacy study (i.e., 50 mg per kg), and compared this with both vancomycin and vehicle control. As shown in Fig. 4a, gut microbiota diversity in NEN-treated and control mice are only slightly but statistically significant different on day 3, however, the difference is minimal compared to the large reduction observed after vancomycin treatment. This small difference disappears on day 6, while that observed in the vancomycin-treated group remains high. Further, ordination analysis shows the minimal effect of NEN and the large effect of vancomycin on the gut microbiota structure at all time points post-treatment compared to controls (Supplementary Figure 5a). Vancomycin treatment dramatically lowered the diversity of the microbiota, shifting the composition to high relative abundance of *Lactobacillaceae* and *Enterobacteriaceae* (Fig. 4a) as seen previously[38]. Thus, we conclude that NEN treatment has a minimal effect on the gut microbiota.

Next, we evaluated the effects of NEN on the gut microbiota in *C. difficile*-infected mice that had been pre-treated with an antibiotic cocktail 6, 5, and 4 days prior to infection and clindamycin the day prior to infection (Fig. 4b–d). The diversity of the microbiota following NEN treatment (50 mg per kg) on days 1, 2, and 3 post-infection was indistinguishable from the water control group (Fig. 4b). Further, the composition and structure of the gut microbiota on day 6 post-infection did not differ from water treatment (Fig. 4b and Supplementary Figure 5), comprising of high relative abundance of *Lactobacillaceae*, *Bifidobacteriaceae*, *Clostridiales*, and *Bacteroidales*, and decreased relative abundance of *Enterobacteriaceae* (mainly *Escherichia coli*) (Supplementary Figure 5 and Supplementary Figure 6). Linear discriminant analysis (LDA) effect size (LEfSe) analysis[39] only identified members of the genus *Bacteriodes* out of 154 phylotypes as significantly more abundant on day 6 in water control than in NEN treated mice (Supplementary Figure 7).

Having shown that NEN does not affect the structure or composition of the microbiota, we next asked whether NEN (alone or combination with vancomycin) was potentially able to help restore the gut microbiota during the resolution phase of infection. To this end, we compared the effects of NEN (50 mg per kg) + vancomycin (0.5 mg per kg), and NEN (50 mg per kg) alone, to vancomycin alone, in the recurrent *C. difficile* model. As expected, the diversity of the gut microbiota after vancomycin treatment alone remained low throughout the treatment cycle and was dominated by *Lactobacillaceae* (Fig. 4c, d). Whereas the addition of NEN (50 mg per kg) to vancomycin (0.5 mg per kg) showed no benefits to the microbiota compared to vancomycin alone (Fig. 4c), treatment with NEN (50 mg per kg) alone resulted in a significant increase in diversity post-treatment and during resolution of infection (Fig. 4d), indicating that NEN may have additional benefits on the gut microbiota as a stand-alone therapy.

## Discussion

The global spread of epidemic strains of *C. difficile* capable of causing outbreaks and life-threatening infections is a recent phenomenon that has been brought on, in part, by modern human practices. The widespread introduction of the food additive trehalose, shortly before the emergence of epidemic RT027 and RT078, has been proposed to have contributed to selecting for these strains and increasing their virulence[7]. Similarly, the use and misuse of antibiotics have further accelerated the enrichment of multidrug-resistant variants of *C. difficile*, whilst disrupting the protective microbiota that normally prevents such infections[40]. As a result, *C. difficile* continues to increasingly become more widespread, more virulent, and more difficult to treat with traditional eradication approaches (i.e., antibiotics). The notion of targeting the virulence determinants of *C. difficile* has emerged as an attractive alternative strategy to treat CDI[30,41–43], especially given the role that toxins play in all aspects of disease pathogenesis[9–11,44]. The recent clinical demonstration of disease recurrence attenuation by the injectable TcdB-targeted antibody bezlotoxumab[15] supports these approaches and has fueled efforts to identify next generation antivirulence therapeutics. In particular, more convenient oral agents (i.e., small molecules) that can be dosed at all stages of disease are highly sought after. Moreover, although targeting TcdB appears to be capable of decreasing recurrence, blocking the actions of TcdA and binary toxin, both of which contribute to disease pathogenesis in hypervirulent strains and appear to be sufficient for causing disease in strains lacking TcdB in certain cases, would be a highly desirable feature of any comprehensive would-be antivirulence strategy.

In this study, we performed a high-throughput screen of libraries containing FDA- and EMEA-approved drugs to identify small-molecules that protected cells from TcdB intoxication that could potentially be repositioned as orally-bioavailable therapeutics for treating CDI. Among the dozens of hits identified in the primary screen (Supplementary Figure 1), we noted that several were approved anti-parasitic drugs. From the most potent inhibitors of cell-rounding in this class, niclosamide was selected for further characterization based on its impeccable safety profile in humans, and known preferential distribution in the lower GI after oral dosing[26], which we anticipated would be beneficial for targeting the gut-damaging toxins of *C. difficile*. Niclosamide is a remarkably well-studied molecule that has been shown to have a number of other biological activities in vitro that have prompted other investigations into translation into diseases including cancer[45–49], diabetes[50] as well as other infectious diseases[22]. In most cases, however, the low systemic exposure of niclosamide,

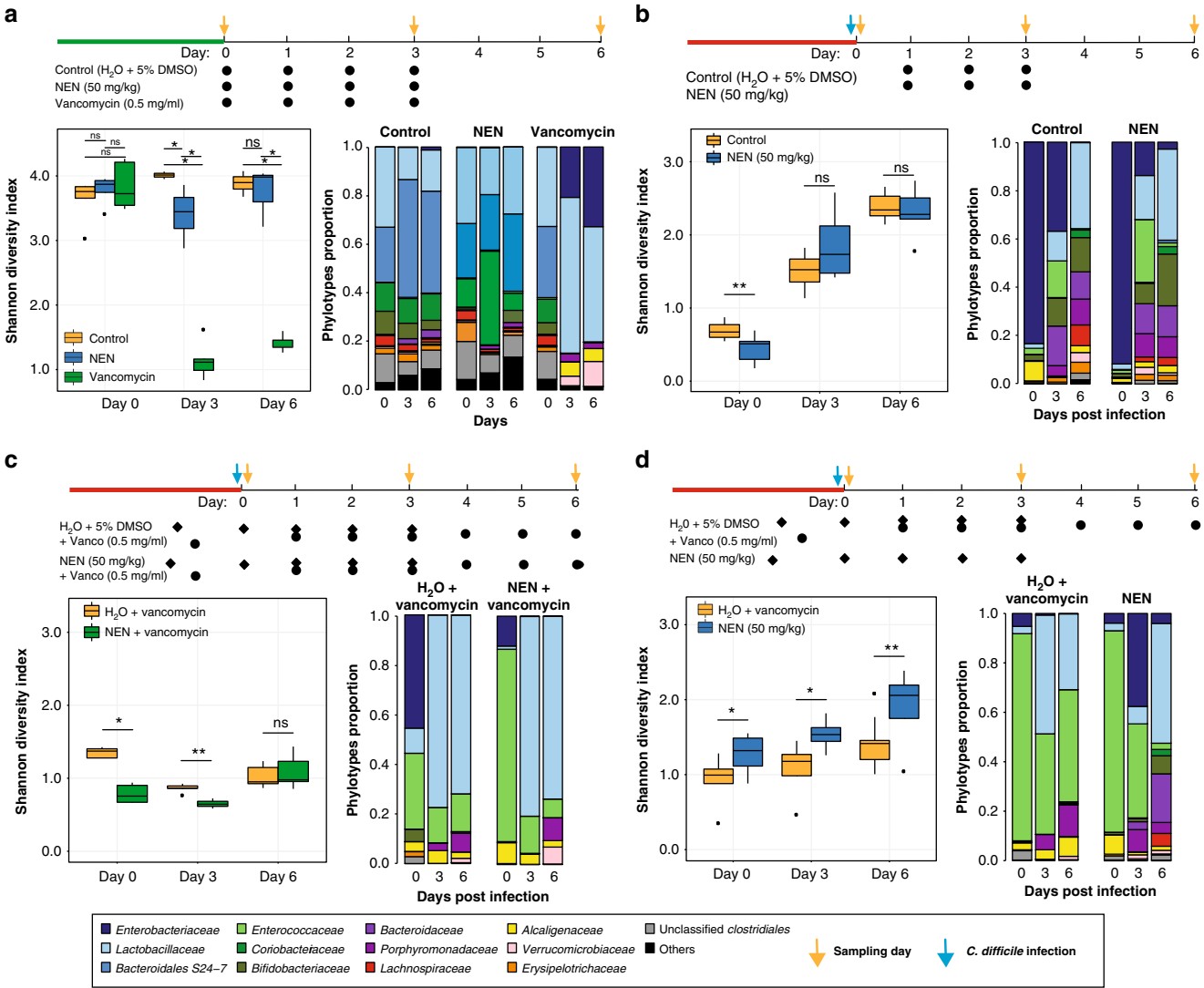

**Fig. 4** Effect of treatment on the gut microbiota diversity and composition. Experimental design is shown on top panel (red line indicates antibiotic cocktail treatment 6 days prior to the *C. difficile* infection for 3 days, followed by 2 days of water and clindamycin (30 mg per kg) the day prior to *C. difficile* infection, green line indicates no treatment). Within-sample diversity was estimated using Shannon diversity index (ns denotes not significant, *p value greater than 0.01 but not greater than 0.05, **p value is less than 0.01, unpaired Student *t*-test) The center line, bounds of box, and whiskers of the boxplot represents median, interquartile, and range of within-sample diversity. Each treatment group is the result of two experimental cages. Each barplot indicates the mean relative abundance of bacterial families with relative abundance >1% from mice in two experimental cages. The *Peptostreptococcaceae* family of bacteria includes *C. difficile* and its abundance was consistently below 1% in all experiments and thus is included in others. **a** Control, NEN, and vancomycin treatment in the absence of *C. difficile* infection; **b** control (water with 5% DSMO) and NEN treatment (50 mg per kg); **c** water with vancomycin and NEN treatment (50 mg per kg); **d** water with vancomycin and NEN treatment (50 mg per mL) with vancomycin (0.5 mg per mL). First sampling day (day 0) is 4 h after *C. difficile* infection

which is likely a major contributor to its overall safety, has hampered its use for indications outside of the GI tract. Nevertheless, efforts have been undertaken to improve the bioavailability of niclosamide, using different salt forms, chemical modifications, and the use of nanoparticles[51–53].

Here, we found that niclosamide provided protection to a variety of human cells from both TcdB-induced cell-rounding, and TcdB-induced necrosis. Moreover, due to its mechanism of inhibition of TcdB (i.e., targeting host endosomal pH through a proton-shuttle mechanism), niclosamide had the added major benefit of being equally effective against TcdA and CDT—both of which also require low pH for entry into the host. These features provided the unique opportunity to test niclosamide as a stand-alone agent against epidemic strains of *C. difficile* that are triple-

positive for TcdA, TcdB, and CDT. In mouse models of CDI, we showed that NEN dose-dependently improved disease symptoms associated with both primary infection and recurrence, with full protection seen at 50 mg per kg NEN. Of note, the reported median oral lethal dose ($LD_{50}$) for NEN in rats is 10,000 mg per kg body weight[27], further emphasizing the large therapeutic index for NEN in treating CDI. Finally, an important and significant finding in this study was that NEN had no major deleterious effects on the structure and composition of the microbiota (Fig. 4). Increased diversity of the gut microbiota was observed after NEN treatment, a feature that was thought to be unique to fecal microbiota transplantation and key to its success. Recovery of a diverse microbiota might in part contribute to its curative properties. Beyond this study and this indication, the findings

presented here pave the way to future studies aimed at investigating the effects of NEN on other toxin-driven enteric diseases. Also, given the increasingly appreciated role that helminths play in interacting with and modulating the microbiome[54,55], the results here highlight the importance of future studies aimed at understanding the effects of NEN on the microbiota during a helminth infection.

## Methods

**Cell lines, consumables, and reagents.** Plasticware used for cell culture and enzyme assays were purchased from Corning. Streptavidin Hi-bind plates, Superblock buffer, SuperSignal West, and Quantablu peroxidase substrate were purchased from Thermo Pierce (Rockford, IL). Cell lines CHO-K1 (ATCC, Cat #CCL-61), Vero (ATCC, Cat #CCL-81), CaCo-2 (ATCC, Cat #HTB-37), HCT-116 (ATCC, Cat #CCL-247), and IMR90 (ATCC, Cat #CCL-186) were purchased from ATCC (Manassas, VA). Anti-TcdB antibody (used at 1:2000 dilution) was purchased from R&D Systems (Cat #AF6246). Anti-Rac1 antibody Mab102 (used at 1:1000 dilution) was purchased from BD Biosciences (Mississauga, ON), Cat #610651. Anti-Rac1 antibody 23A8 (used at 1:1000 dilution) was purchased from Millipore (Cat #05-389). Anti-ERK and Anti-EEA1 (used at 1:2000 dilution) were purchased from Cell Signalling Technologies (Cat #137F5) and Millipore (Cat #07-182-25UL), respectively. Anti-mouse conjugated peroxidase antibody (used at 1:10,000 dilution) was purchased from GE Healthcare (Cat #NXA931V). Anti-rabbit HRP (used at 1:10,000 dilution) was purchased from GE Healthcare (Cat #NA934V). Anti-sheep HRP (used at 1:100,000 dilution) was purchased from Invitrogen (Cat #81-8620). The Spectrum library, consisting of 2320 individual compounds formatted as 10 mM solutions in DMSO, was purchased from Microsource (Gaylordsville, CT). The LOPAC[1280] was purchased from Sigma-Aldrich (Oakville, ON).

**Protein expression and purification.** Expression and isolation of recombinant toxins were as described by Yang et al.[56]. Briefly, transformed *Bacillus megaterium* was inoculated into LB containing tetracycline and grown to an A600 of 0.7, followed by overnight xylose induction at 37 °C. Bacterial pellets were collected, resuspended with 20 mM Tris pH 8/0.5 M NaCl, and passed twice through an EmulsiFlex C3 microfluidizer (Avestin, Ottawa, ON) at 15,000 psi, then clarified by centrifugation at 18,000*g* for 20 min. TcdB was purified by nickel affinity chromatography followed by anion exchange chromatography using HisTrap FF and HiTrap Q columns (GE Healthcare, Baie D'Urfe, QC), respectively. Fractions containing TcdB were verified by SDS-PAGE, then pooled and diafiltered with a 100,000 MWCO ultrafiltration device (Corning) into 20 mM Tris pH 7.5/150 mM NaCl. Finally, glycerol was added to 15% v/v, the protein concentration was estimated by A280 (using coefficient 288160), divided into single-use aliquots, and stored at −80 °C.

For TcdA, cell lysates were prepared as described for full-length TcdB, and purification of the protein was by nickel affinity chromatography using HisTrap FF columns.

The pGEX-Rac1 plasmid (Addgene plasmid 12200) for expression of GST-Rac1 protein was previously described by Bagrodia et al.[57] and obtained from Addgene (Cambridge, MA). The plasmid was transformed into *E. coli* BL21 DE3, and recombinant protein expression was achieved by induction of the culture with 0.1 mM IPTG for 5 h at 30 °C. The cell pellet was recovered by centrifugation, resuspended with 5 mL/g of pellet in 20 mM Tris pH 7.5/150 mM NaCl, and sonicated. The cell lysate was clarified by centrifugation, and the GST fusion protein was purified by chromatography through a GSTrap Fast Flow column (GE Healthcare, Baie D'Urfe, QC). Following elution with 10 mM glutathione, fractions containing purified protein were pooled and stored at −80 °C in the presence of 15% v/v glycerol.

**Arrayscan high content phenotypic screen.** IMR90 cells were grown in EMEM (Wisent) supplemented with 10% FBS and penicillin–streptomycin (complete EMEM) and were seeded in 96-well Cellbind plates (Corning) at a density of 8000–10,000 cells/well. The next day, the media was exchanged with serum-free EMEM (SFM) containing 1 μM CellTracker Orange CMRA (Molecular Probes). After 60 min, excess dye was removed by media exchange with SFM. An Agilent Bravo liquid handler was used to deliver 0.4 μL of compound from the Microsource library plate to the cell plate, immediately followed by 10 μL of 100 pM TcdB (diluted in SFM) to a final volume of 100 μL, representing a concentration of toxin previously established as ~EC99 levels of cytopathology. The cell plates were returned to the incubator for 3.5 h before imaging. CellTracker-labeled cells were evaluated on a Cellomics ArrayScan VTI HCS reader (Thermo Scientific, Waltham, MA) using the Target Acquisition mode, a 10× objective and a sample rate of 100 objects per well. After recording all image data, the cell rounding and shrinking effects of TcdB intoxication were calculated using the cell rounding index (CRI), a combined measure of the length to width ratio (LWR) and area parameters. The % inhibition was calculated as the ratio between the sample well and the average toxin-untreated controls after subtracting the average DMSO control values. The Z′ value was calculated using the equation $Z' = 1 - [(3s_f + 3s_b)/(\mu_b - \mu_f)]$, where $s$ =

standard deviation, $\mu$ = average, $f$ = DMSO control, and $b$ = toxin-untreated control. Wells which displayed potential suppression of toxin activity (>39%) were verified by visual inspection to immediately exclude false hits arising from cellular toxicity, precipitation, or auto-fluorescence/quenching. Hits for confirmation and follow-up assays were ordered from Microsource and Sigma as lyophilized powders. Dose response curves were created and evaluated using Prism software (GraphPad Software, La Jolla, CA).

**Acute toxicity assay (necrosis).** Loss of cellular ATP as a marker of high dose (1 nM) TcdB toxicity was measured as described for the Arrayscan screen protocol, except that CellTiter-Glo reagent (Promega, Madison, WI) was added to the cells 3-h post toxin challenge, and luminescence was recorded on a Spectramax M5 plate reader.

**LysoTracker assay.** Endosomal pH neutralization was assayed essentially as described by Slater et al.[58]; IMR90 cells in complete EMEM were plated at 14,000 cells/well (~95% confluency). After 24 h, the media was changed to SFM for 60 min, then compound was added to 40 μM and incubated at 37 °C for 2 h. LysoTracker red DND-99 and Hoechst (Life Technologies) were added to 0.1 and 1 μM, respectively, and incubated for 60 min. Excess dye was removed by media change and the fluorescence was read at ex/em 574/594 was read on an Envision plate reader (Perkin Elmer). Representative cell images were taken using a Zeiss Axiovert fluorescence microscope using DAPI and Texas Red filters to visualize the Hoechst and LysoTracker staining, respectively.

**CDT toxicity assay.** Vero cells were plated in 96-well Cellbind plates at a density of 8000 cells/well (~90% confluent). The next day, compound dilutions and CDT binary toxins (2 μg/mL each of A and trypsin-activated B—a kind gift from Merck) in serum-free DMEM were added to the plate. After 24 h, wells were washed with PBS, and the cells were fixed with 4% paraformaldehyde for 15 min, permeabilized with 0.25% TX100 for 5 min, then F-actin was stained with Phalloidin Alexa488 (Thermo) for 2 h before washing and reading Alexa488 fluorescence on a Molecular Devices Spectramax M5e (bottom read, well scan, 9 points/well).

For photomicroscopy, the cells were stained with 1 μM Hoechst for 30 min, then combined photos were taken for each compound at a concentration corresponding to maximum protection from TcdB using a 10× objective and appropriate filter sets for Hoechst and Alexa488 fluorescence (Zeiss).

**TcdB in vitro autoprocessing assay.** Inhibition of TcdB self-cleavage by its intrinsic cysteine protease activity was measured by pre-incubating test compounds with TcdB for 30 min, followed by addition of InsP6 and incubating the reaction at 37 °C for 3 h. Cleavage was visualized by electrophoresing the samples on SDS polyacrylamide gels and staining with Coomassie Blue R250.

**Rac1 glucosylation.** IMR90 cells were grown in 6-well plates at a density of 300,000 cells/well. The next day, TcdB (0.5 pM final) and NEN (2A pharmachem) were added to the wells in serum-free media. After 60 min the cells were harvested and lysed in Laemmli buffer. SDS PAGE and western blotting were performed to detect glucosylated Rac1 using mAb102 (BD Biosciences), total Rac1 using mAb23A8 (Abcam), and loading controls using anti-EEA1 and anti-ERK1/2 (EMD Millipore). Raw Western Blot images are shown in Supplementary Figure 8.

**Differential scanning fluorometry.** Differential scanning fluorometry (DSF) was performed in a similar manner as described previously[59]. TcdB protein was diluted in phosphate buffer (100 mM KPO₄, 150 mM NaCl, pH 7) containing 5× SYPRO Orange (Invitrogen, Burlington, ON) and a serial dilution of test compound. A BioRad CFX96 qRT-PCR thermocycler was used to establish a temperature gradient from 15 °C to 95 °C in 30 s increments, while simultaneously recording the increase in SYPRO Orange fluorescence as a consequence of binding to hydrophobic regions exposed on unfolded proteins. The Bio-Rad CFX Manager 3.1 software was used to integrate the fluorescence curves to calculate the melting point.

**TcdB cell surface binding.** HCT116 cells in 10 cm dishes were grown to 90% confluence. TcdB (2 nM) and either 100 μM methylcholate or 1 μM NEN were preincubated together for 30 min on ice in serum-free media before adding to cells. After incubating for 60 min on ice, cells were washed with PBS, harvested and lysed in 300 μL of 0.5% TX100/PBS. Clarified material was analyzed by western blot by probing with anti-TcdB antibody (R&D Systems AF6246) and anti-tubulin antibody as a loading control. TcdB bands were measured by densitometry using a ChemiDoc MP Imaging System (BioRad).

**UDP-Glo™ UDP-glucose hydrolase assay (Promega).** Experiments were performed as per the manufacturer's instructions. Briefly, 100 nM of GTD enzyme was incubated in glucosylation buffer (see above) with various concentrations of inhibitor in a final volume of 16 μL. Reactions were started with the addition of 4 μL of UDP-glucose (50 μM final). Reactions were allowed to proceed at room

temperature for 15 min. To stop the reaction, 10 μL were removed and added to a white, polystyrene 96-well half-area plate (Costar) containing 10 μL of UDP detection reagent. Plates were incubated at room temperature for 1 h, then luminescence was recorded on a SpectraMax M5e plate reader (Molecular Devices) with an integration time of 750 ms. Results were analyzed with SoftMax Pro 6.2.2 and GraphPad Prism 5.0.

**MIC assays to determine effects of NEN on *C. difficile* strains**. For results shown in Table 2: Minimal Inhibitory Concentration (MIC) testing was performed by the National Research Council of Canada (Ottawa), using the following bacterial strains: *Clostridium difficile* R20291 (UK) O27 ribotype, *Clostridium difficile* M68[60] 017 ribotype, *Clostridium clostridioforme* ATCC25537, and *Clostridium sporogenes* ATCC3584. Compounds were dissolved in DMSO and added to Brucella supplemented blood plates. Niclosamide and NEN from a 10 mM stock solution in DMSO, was tested at 50 μM starting concentration (16 and 19 μg/mL, respectively), with 1 in 3 serial dilutions. Vancomycin was tested as a positive control starting at 8 μg/mL, with 1 in 2 serial dilutions.

Inoculum preparation: strains were grown on Brucella supplemented blood agar plates in an anaerobic chamber for 16 h. For each strain, cells were harvested into saline to OD 0.1. The CFU/mL was confirmed by serial dilution and plating onto Brucella supplemented blood or Braziers agar for each experimental group.

For MIC Testing, 10 μL of each strain was spotted three times onto each agar plate and incubated at 37 °C in an anaerobic chamber. Each plate was examined for visible growth at 16, 24, 32, and 48 h. No further change in growth pattern was observed for any strain from 16 to 48 h growth. MIC determinations were performed in duplicate and reported as the concentration that completely inhibits growth.

For results represented in Table 3: Minimal Inhibitory Concentration (MIC) testing was performed by Micromyx. NEN and comparator antibiotics were prepared on the day of testing using solvents recommended by CLSI. Stock solutions of all compounds were made at 100× the final testing concentration. Test organisms consisted of clinical isolates from the American Type Culture Collection (ATCC) and Micromyx repository. Drug dilutions and drug-supplemented agar plates were prepared manually. After pouring the Supplemented Brucella agar plates, they were allowed to dry, pre-reduced in the Bactron II anaerobic chamber, then spot-inoculated using a Steers Replicator, yielding a final cell concentration on the surface of the agar plates of ~1 × 10^4 colony-forming units/spot. After the inocula had dried, the drug-supplemented plates were incubated at 35 °C for 16, 24, 32, and 48 h under anaerobic conditions. The MIC was read per CLSI guidelines as the concentration at which growth was significantly inhibited relative to the growth control.

**In vivo studies**. All procedures involving animals were conducted under protocols approved by the Institutional Animal Care and Use Committee at the University of Maryland, Baltimore.

For the primary CDI Model[61–64], C57BL/6 mice (10 per group) were orally administered 10^5 CFU of *C. difficile* spores from the UK1 (BI/NAP1/027) strain after receiving antibiotic treatment in the drinking water for 3 days, as shown in Fig. 3. The antibiotic mixture contains: kanamycin (0.4 mg per mL), gentamicin (0.035 mg per mL), colistin (850 U per mL), metronidazole (0.215 mg per mL), and vancomycin (0.045 mg per mL) was prepared. This corresponds to the approximate daily dose used for each antibiotic such as kanamycin (40 mg per kg), gentamicin (3.5 mg per kg), colistin (4.2 mg per kg), metronidazole (21.5 mg per kg), and vancomycin (4.5 mg per kg). Mice were given an intraperitoneal injection of clindamycin (10 mg per kg) 1 day before spores challenge. Mouse weights and the development of disease symptoms were monitored daily. Animals that became moribund or lost >20% of their body weight were euthanized. The mice were divided into the following groups: Control (water with 5% DMSO), NEN (2, 10, or 50 mg per kg suspended in water with 5% DMSO)

For the recurrent CDI Model, C57BL/6 mice (10 per group) were prepared for primary CDI model as previously mentioned; however, mice were orally administered vancomycin (0.45 mg per mL) in the drinking water for 7 days post spores challenge. Mice were given regular water for until the end of the study. Mouse weights and the development of disease symptoms were monitored daily. Animals that became moribund or lost >20% of their body weight were euthanized. Diarrhea was scored as following: 0 (normal hard fecal pellet); 1 (hard to produce fecal pellet yet no rectal inflammation); 2 (liquid feces, inflamed rectum, soiled tail).

**Microbiota analysis**. DNA was extracted using the MagAttract PowerMicrobiome DNA/RNA kit (Qiagen) from the fecal pellet of all samples. Briefly, the glass bead plate was used to mix fecal material and lysis solution, and inhibitor was subsequently removed from the supernatant. ClearMag Beads suspension was then mixed with 450 μL of the supernatant to purify the extracted DNA.

DNA was extracted from all samples (150 mg) using the MagAttract PowerMicrobiome DNA/RNA kit (Qiagen) implemented on a Hamilton STAR robotic platform and after a bead-beating step on a TissueLyzer II (Qiagen) in 96-deep well plates. PCR amplification of the 16S rRNA gene V4 hypervariable region was performed using dual-barcoded universal primers 515F (GTGYCAGCMGCCGCGGTAA) and 806R (GGACTACNVGGGTWTCTAAT)

as previously described[65]. High-throughput sequencing of the amplicons was performed on an Illumina MiSeq platform using the 300 bp paired-end protocol. Raw data was demultiplexed using the idemp tool[66]. Barcode, adapter, and primer sequences were trimmed using tagcleaner[67]. Quality assessment and sequencing error correction was performed using the software package DADA2[68] and the following parameters: forward reads were truncated at position 220 and the reverse reads at position 160 based on the sequencing quality plot, no ambiguous based and a maximum of 2 expected errors per-read were allowed[69]. The quality-trimmed reads were used to infer ribosomal sequence variants and their relative abundance in each sample after removing chimera. A total of 205 fecal samples were characterized resulting in a total of 6,619,465 high-quality non-chimeric amplicon sequences, corresponding to 32,290 (±19,999) sequences per samples. The GreenGene database version 13.8[70] was used for taxonomic classification. Within-sample diversity was estimated using Shannon diversity index[71]. Inter-community comparative analyses were performed using NMDS (nonmetric dimensional scaling) based on Bray–Curtis distance metrics and were plotted using software package phyloseq[72]. Linear discriminant analysis (LDA) effect size (LEfSe) analysis[39] was applied to identify bacterial phylotypes with relative abundance statistically different between control and NEN (50 mg per kg) treatments. The alpha value for the non-parametric factorial Kruskal–Wallis (KW) sum-rank test[73] was 0.05 and the threshold for the logarithmic LDA model[74] score for discriminative features was set at 2.0.

## Data availability
All 16S rRNA sequence data generated and analyzed during the current study were deposited and are available in SRA under BioProject PRJNA423011 (SRP128045).

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

## Acknowledgements

This work was supported by a Canadian Institutes of Health Research Industry-Partnered Collaborative Research Operating Grant (to R.A.M.). The authors thank

Merck Research Laboratories for generous support in kind of reagents and materials. The authors also wish to thank Chris Fladd from the SPARC BioCentre at The Hospital for Sick Children for helpful assistance in setting up and developing the imaging-based screening platform, and Dr. Nachum Kaplan and Owen Roberts for their contributions towards studies related to evaluating the antimicrobial properties of niclosamide.

## Author contributions

J.T., T.H., H.F., and R.A.M. designed the research. J.T., T.H., B.M., K.C., and G.L.B. performed the research. J.T., T.H., B.M., K.C., G.L.B., J.R., H.F., and R.A.M analyzed the data and R.A.M. wrote the paper.

## Additional information

**Competing interests:** The authors declare no competing interests.

