## [Peer Review File · Nature Communications]

Reviewers' comments:

Reviewer #1 (Remarks to the Author):

In their manuscript entitled "Oral niclosamide inhibits *C. difficile* virulence and prevents disease pathogenesis in mice without disrupting the gut microbiota" by Tam et al. they use a high throughput approach to screen approved FDA drugs that are able to inhibit *C. diff* toxins. The team found multiple drugs that were able to inhibit toxin, and rationally selected niclosamide for further studies to systematically define the mechanism of action. They go on to show in a very elegant set of experiments that this drug is able to interact with the host; through inhibition of a pore formation process and is able to neutralize TcdB activity. They also show that this drug blocks the activity of TcdA and CDT toxin, which is a significant finding. This section of the manuscript is well done and very exciting. They then ask if this drug is able to alter the gut microbiota, and conduct a series of mouse models concluding that niclosamide or NEN is able to treat CDI, and also prevent recurrence in a mouse model by restoring the gut microbiota after treatment compared to vancomycin treatment.

Most of the conclusions in this manuscript are supported by the data as far as the toxin MOA studies. However, the mouse model studies paired with the microbiota data in Figures 3 and 4 need to be clarified further. The methods section is missing how the mouse model experiments were done in Figure 3 and 4. This does not diminish the major findings of this paper where the authors found a drug that is able to neutralize all three toxins made by *C. difficile*. This is novel, significant, and will be well received by the wider scientific community. The approach that the authors took to find this drug is also very innovative. The experiment and statistical approaches look appropriate and robust. Again, a large section is missing from the methods at this time.

Major comments

Most of my comments center around the work done in Figure 3, 4, and the methods section.

1. The title for the paper states that this drug inhibits *C. difficile* virulence and prevents disease pathogenesis in the mouse model. The authors do not show disease in the mouse model only score diarrhea, weight loss, and death. Death is definitely a metric of severe disease, but since this paper is about a drug that can neutralize toxin, I think looking at toxin or damage from the toxin in vivo would have been a priority. Histopathological changes to the cecum and colon with and without NEN and vanco treatment should be a priority to show that the toxin is neutralized in vivo.
2. In the prevention mouse model in Figure 3 (challenge after 4hr spore gavage), is NEN in water the control or should this be in DMSO? What is NEN resuspended in and if it is not water, should this be the vehicle control as it is shown in Figure 4?
3. There are no methods on how the mouse model was done or how diarrhea was scored? I think the authors forgot to add this to the manuscript.
4. The recurrence model that is presented in Figure 3E-H is not really a recurrence model as the authors do not look at disease in the mice before they start treatment with vanco or NEN. Recurrence or relapse occurs when patients have disease from CDI, significant inflammation, and then are treated with vanco. The disease resolves and then it comes back again worse than before. There are better mouse models of recurrence. I think this is a model that is looking NEN in concert with vancomycin. Even with treatment the mice still lose weight at day 2 and 11. It would be helpful to know what the *C. difficile* bacterial load was at this point, and also disease via inflammation in the large intestine. If NEN resolves inflammation in vivo then this is again another significant finding and supports the in vitro work in Fig 1 and 2.

5. The experiments in Figure 4A are a really nice addition. It is really important to see how NEN affects the gut microbiota alone. In this experiment there is a vehicle control. Was this used in the Figure 3 infection studies or was water used? Can you clarify this?

6. It appears that the vehicle control changes the diversity at least on Day 3, and in the paper it says NEN did not change the diversity. In Figure 4A Day 3, it does?

7. In the text it needs to clearly spell out if it is a water or vehicle control group. Since these mouse models are very complex, it would help to have this consistent in the text and the figure. Also, the details in the methods section would be helpful with this.

8. It appears that the starting microbiota for most of the groups are different at on Day 0. One limitation using this antibiotic treated mouse model is that it does not change the microbiota in a reproducible way, so it is hard to compare across different cages from different treatments, even the same treatments. Figure 4 shows this nicely in all Day 0 time points, and Figure S6 shows this clearly as there are clearly cage effects. This makes the interpretations in this section much harder to make. If the authors want to show that the NEN, is better then the vanco then why do they not just have one figure comparing the vehicle, NEN alone, and vanco alone across each treatment? Then look at the microbiota and ordination to show that the NEN treatment helps the microbiota recover faster or differently then vanco. This seems to be a major conclusion and is not supported by the data currently.

9. Where is *C. difficile* in the bar plots? Shouldn't the mice be colonized with it? Please point this out to the reader in the text.

Minor comments

Figure S7 is not very clear to the reader.

Since this drug is important for helminths and there are studies looking at helminths and the microbiota, I think a sentence in the conclusions putting this into context would be helpful.

Reviewer #2 (Remarks to the Author):

The authors report on the discovery of compounds, which block the toxic activity of *C. difficile* toxins A (TcdA) and B (TcdB). By screening of compounds, which are already used as drugs they identified niclosamide and 2 other related molecules as substances, which inhibited the cytopathic (cytotoxic) effects of TcdB. Niclosamide is an approved drug for worm diseases, which predominantly acts in the gut, because systemic up-take is minimal. They showed that niclosamide has no effect on the glucosylation of Rac protein or on toxin processing by inbuilt cysteine protease activity. Moreover, they report that niclosamide does not directly interact with TcdB or block toxin binding to cells. In contrast, they propose that niclosamide inhibits the acidification of endosomes, which is a prerequisite for toxin translocation into the cytosol of host cells. Not only cell up-take of TcdB and TcdA are blocked but also of CDT, which acts by a completely different toxin mechanism. They show that niclosamide or its more soluble preparation NEN protects mice towards *C. difficile*-induced pathogenesis. Moreover, they show that niclosamide does not destroy the normal microbiota as it is known (shown) for antibiotic therapy of *C. diff*-infection e.g. with vancomycin. The finding of the potent action of niclosamide is of some interest. However, the precise mechanism of action of niclosamide has not been studied and clarified. Moreover, the paper contains several unclear statements.

Specific comments

1. I miss time curves of the intoxication of cells. In some experiments it seems that niclosamide blocks completely the effect of TcdB (e.g., Fig. 1f and g). However, one needs to have time courses of the intoxication process to see this in detail.
2. Intoxication of cells was performed in serum free medium. This enhances generally the toxin activity. However, serum might completely block the effect of niclosamide. Therefore, also these conditions are important.
3. The picture in Fig. 1e is much too small and bars are missing.
4. In general, the Fig. legends are much too short.
5. Fig. 2f is strange. Why causes the addition of a low amount of TcdB such an increase in labelling by lysotracker. One would expect that also under control conditions without toxin low pH endosomal compartments exist, which should be labeled by lysotracker.
6. The Fig. legend of Fig. 2 is incomplete. Only 2a and 2b are described. The legends for Fig. 2 c, d, e, f and g are missing.
7. How was the necrosis measured in Fig. 2d?
8. The assay for the binding of TcdB is problematic (Fig. 2e), because the data may represent unspecific binding. This part is not to understand, because the legend of Fig. 2e is missing.
9. The quality of Suppl. Fig. 4 is not satisfying.
10. What is the reason for the poor systemic up-take of niclosamide, when it enters cells at low concentrations. Or is this effect on acidification of endosomal compartments only caused by endocytosis?

Reviewers' comments:

Reviewer #1 (Remarks to the Author):

In their manuscript entitled “Oral niclosamide inhibits *C. difficile* virulence and prevents disease pathogenesis in mice without disrupting the gut microbiota” by Tam et al. they use a high throughput approach to screen approved FDA drugs that are able to inhibit *C. diff* toxins. The team found multiple drugs that were able to inhibit toxin, and rationally selected niclosamide for further studies to systematically define the mechanism of action. They go on to show in a very elegant set of experiments that this drug is able to interact with the host; through inhibition of a pore formation process and is able to neutralize TcdB activity. They also show that this drug blocks the activity of TcdA and CDT toxin, which is a significant finding. This section of the manuscript is well done and very exciting. They then ask if this drug is able to alter the gut microbiota, and conduct a series of mouse models concluding that niclosamide or NEN is able to treat CDI, and also prevent recurrence in a mouse model by restoring the gut microbiota after treatment compared to vancomycin treatment.

Most of the conclusions in this manuscript are supported by the data as far as the toxin MOA studies. However, the mouse model studies paired with the microbiota data in Figures 3 and 4 need to be clarified further. The methods section is missing how the mouse model experiments were done in Figure 3 and 4. This does not diminish the major findings of this paper where the authors found a drug that is able to neutralize all three toxins made by *C. difficile*. This is novel, significant, and will be well received by the wider scientific community. The approach that the authors took to find this drug is also very innovative. The experiment and statistical approaches look appropriate and robust. Again, a large section is missing from the methods at this time.

We thank you for your comments. Regarding the missing methods, we sincerely apologize for these not being included in the submitted manuscript. They somehow disappeared from the penultimate draft. In the current draft, a complete methods section is included.

Major comments

Most of my comments center around the work done in Figure 3, 4, and the methods section.

1. The title for the paper states that this drug inhibits *C. difficile* virulence and prevents disease pathogenesis in the mouse model. The authors do not show disease in the mouse model only score diarrhea, weight loss, and death. Death is definitely a metric of severe disease, but since this paper is about a drug that can neutralize toxin, I think looking at toxin or damage from the toxin in vivo would have been a priority. Histopathological changes to the cecum and colon with and without NEN and vanco treatment should be a priority to show that the toxin is neutralized in vivo.

We are very sensitive to the fact that terms like: ‘infection’, ‘disease’, ‘associated disease’, ‘symptoms’ and ‘outcomes’ are often conflated and confused in the literature when it comes to *C. difficile*. We have gone through the manuscript and have made changes in the main text and the title to be more explicit and precise with our descriptions. With respect to the particular experiments used to demonstrate the mechanism of action of prevention of disease by NEN, we feel that the toxin neutralization via NEN was more than sufficiently

demonstrated in this study by the wealth of data generated showing unequivocally that NEN (i) completely inhibited the pathogenesis of all three toxins through a mechanism that is both well validated and well understood; (ii) had no antibacterial effects on *C. difficile* itself; (iii) did not disrupt the structure or composition of the microbiota; (iv) showed attenuation of toxin-dependent diarrhea and weight loss; and, (v) showed a dramatic protection from the most severe outcome of CDI (i.e, death), which depends complete on toxin action.

2. In the prevention mouse model in Figure 3 (challenge after 4hr spore gavage), is NEN in water the control or should this be in DMSO? What is NEN resuspended in and if it is not water, should this be the vehicle control as it is shown in Figure 4?

Thank you for catching this – the control is 5% DMSO, and NEN is given in 5% DMSO. This was added in the figure legend.

3. There are no methods on how the mouse model was done or how diarrhea was scored? I think the authors forgot to add this to the manuscript.

Yes, we did. Apologies. We were citing our previous published methods but agree to include the detail methodologies to help readership. We have added the following to the methods section:

Primary CDI Model (Chen et al, Gastroenterology, 2008; Wang et al, IAI, 2012; Yang et al, JID, 2015; Yang et al, IAI, 2015; Yang et al, Pathog Dis, 2016): C57BL/6 mice (10 per group) were orally administered 10^5 CFU of *C. difficile* spores from the UK1 (BI/NAP1/027) strain after receiving antibiotic treatment in the drinking water for 3 days, as shown in Figure 3. The antibiotic mixture contains: kanamycin (0.4 mg/mL), gentamicin (0.035 mg/mL), colistin (850 U/mL), metronidazole (0.215 mg/mL), and vancomycin (0.045 mg/mL) was prepared. This corresponds to the approximate daily dose used for each antibiotic such as kanamycin (40 mg/kg), gentamicin (3.5 mg/kg), colistin (4.2 mg/kg), metronidazole (21.5 mg/kg), and vancomycin (4.5 mg/kg). Mice were given intraperitoneal injection of clindamycin (10 mg/Kg) one day before spores challenge. Mouse weights and the development of disease symptoms were monitored daily. Animals that became moribund or lost >20% of their body weight were euthanized. The mice were divided into the following groups: Control (water with 5% DMSO), NEN (2, 10, or 50 mg/Kg suspended in water with 5% DMSO)

Recurrent CDI was performed follow our previous established model (Sun et al, IAI, 2011) with modifications: C57BL/6 mice (10 per group) were prepared for primary CDI model as previously mentioned; however, mice were orally administered vancomycin (0.45 mg/mL, starting from day 1 post spore challenge) in the drinking water for 6 days post spores challenge. Mice were given regular water for until the end of the study. Mouse weights and the development of disease symptoms were monitored daily. Animals that became moribund or lost >20% of their body weight were euthanized.

Diarrhea was scored as following: 0 (normal hard fecal pellet); 1 (hard to produce fecal pellet yet no rectal inflammation); 2 (liquid feces, inflamed rectum, soiled tail)

4. The recurrence model that is presented in Figure 3E-H is not really a recurrence model as the authors do not look at disease in the mice before they start treatment with vanco or NEN. Recurrence or relapse occurs when patients have disease from CDI, significant inflammation, and then are treated with vanco. The disease resolves and then it comes back again worse then before. There are better mouse models of recurrence. I think this is a model that is looking NEN in concert with vancomycin. Even with treatment the mice still lose weight at day 2 and 11. It would be helpful to know to know what the *C. difficile* bacterial load was at this point, and also disease via inflammation in the large intestine. If NEN resolves inflammation in vivo then this is again another significant finding and supports the in vitro work in Fig 1 and 2.

We have monitored disease before and during the treatment with vanco (Figure 3F and 3H) and as it showed that the disease symptoms (weight loss and diarrhea) were apparent before we started to the treatment of vancomycin, which allowed us to have consistent recurrent CDI. We did not wait to treat mice when the diseases became the most severe (day 2-3 post challenge) since at that time around 50% mice would be moribund and have to be excluded from the experiments. The schedule of NEN treatment was used as the same as in the primary CDI model since the dose and schedule gave us a consistent reduction of disease severity. The recurrent CDI model used is meant to simulate the clinical setting where patients receive vancomycin treatment after the disease. The disease may come back when patients stop the antibiotic course and disease severity may vary. Herein, mice are given vancomycin 24hrs post spores challenge and evaluate whether NEN will help to reduce the recurrent disease. During the early treatment (day 2) and shortly after the antibiotic withdraw, the disease symptoms were still evident in NEN treatment group, but significantly lower than those in control group.

5. The experiments in Figure 4A are a really nice addition. It is really important to see how NEN affects the gut microbiota alone. In this experiment there is a vehicle control. Was this used in the Figure 3 infection studies or was water used? Can you clarify this?

Yes. The vehicle control is water with 5% DMSO throughout the entire study.

6. It appears that the vehicle control changes the diversity at least on Day 3, and in the paper it says NEN did not change the diversity. In Figure 4A Day 3, it does?

We agree that gut microbiota diversity in NEN-treated and control mice are slightly but statistically significant different on day 3. However, the difference is small in comparison to the effect of Vancomycin on gut microbiota diversity at day 3. The difference between NEN-treated mice and control mice disappeared on day 6, while the differences between these two groups and vancomycin treated mice remain large. Supplementary Figure 5A (Ordination plot) clearly shows the minimal effect of NEN on the gut microbiota structure. We have clarified the sentence in this section, which was changed from:

“As shown in Fig. 4a, NEN treatment did not affect the high diversity, composition, or structure (Supplementary Fig. 5) of the gut microbiota compared to vehicle control, whereas vancomycin treatment dramatically lowered the diversity of the microbiota, shifting the composition to high relative abundance of *Lactobacillaceae* and *Enterobacteriaceae* (Fig. 4a) as seen previously³⁷.”

to:

“As shown in **Fig. 4a**, gut microbiota diversity in NEN-treated and control mice are only slightly but statistically significant different on day 3, however, the difference is minimal compared to the large reduction observed after vancomycin treatment. This small difference disappears on day 6, while that observed in the vancomycin-treated group remains high. Further, ordination analysis shows the minimal effect of NEN and the large effect of vancomycin on the gut microbiota structure at all time points post-treatment compared to controls (**Supplemental Fig. 5A**). Vancomycin treatment dramatically lowered the diversity of the microbiota, shifting the composition to high relative abundance of *Lactobacillaceae* and *Enterobacteriaceae* (**Fig. 4a**) as seen previously³⁷. Thus, we conclude that NEN treatment has a minimal effect on the gut microbiota.”

7. In the text it needs to clearly spell out if it is a water or vehicle control group. Since these mouse models are very complex, it would help to have this consistent in the text and the figure. Also, the details in the methods section would be helpful with this.

Fair point. This was added to the figure legend in Figure 3. Mouse infection models were added to the Materials & Methods section

8. It appears that the starting microbiota for most of the groups are different at on Day 0. One limitation using this antibiotic treated mouse model is that it does not change the microbiota in a reproducible way, so it is hard to compare across different cages from different treatments, even the same treatments. Figure 4 shows this nicely in all Day 0 time points, and Figure S6 shows this clearly as there are clearly cage effects. This makes the interpretations in this section much harder to make. If the authors want to show that the NEN, is better than the vanco then why do they not just have one figure comparing the vehicle, NEN alone, and vanco alone across each treatment? Then look at the microbiota and ordination to show that the NEN treatment helps the microbiota recover faster or differently than vanco. This seems to be a major conclusion and is not supported by the data currently.

The reviewer correctly points out the limitation of the model, which is characterized by a high variation on day 0 between experiment due to cage effect (see Supplemental Fig. 6). It is true that this is an inarguable inherent feature of this antibiotic treated mice model system. However, this model system is currently the most applicable model system in translational microbiome research and was recommended for studies that aim to investigate the effect of disruption on the gut microbiota (Gut Microbes. 2016; 7(1): 68–74. doi: 10.1080/19490976.2015.1127463). That said, the conclusions drawn from each independent experiment remain valid as within experiment results are consistent. For example, the recovery of the gut microbiota diversity in NEN treated mice is observed after the antibiotic pre-treatment and in the absence of vancomycin treatment and that independently from the composition of the gut microbiota on Day 0. Conversely, vancomycin consistently, reduces gut microbiota diversity which does not recover on day 6. These findings are supported by ordination analysis as shown on Supplemental Figure 6. Thus, our conclusion that NEN treatment contributes to the recovery of the gut microbiota compared to vancomycin treatment, which does not, remains correct.

9. Where is *C. difficile* in the bar plots? Shouldn't the mice be colonized with it? Please point this out to the reader in the text.

For clarity reasons, we have only displayed the family-level taxonomic group that were present at 1% or above. *C. difficile* belongs to the family *Peptostreptococcaceae* which is detected in proportion >1%, thus are included in the “other” category. We did not expect to detect *C. difficile* in feces on Day 0, as sampling is performed right after infection in all the experimental setups. Further, in all treated mice with NEN or vancomycin, it is unlikely that *C. difficile* survived the treatments and thus it would not be detected on days 3 or 6. In untreated controls, the mice do clear the infection which would limit our ability to detect *C. difficile*. It is important to note that we do detect *C. difficile* but in abundance that are less than 1%. We have modified part of the legend of Fig. 4 which now reads:

“... Each barplot indicates the mean relative abundance of bacterial families with relative abundance >1% from mice in two experimental cages. The *Peptostreptococcaceae* family of bacteria includes *C. difficile* and its abundance was consistently below 1% in all experiment and thus is included in “others”. ...”

Minor comments

Figure S7 is not very clear to the reader.

We have included a detailed figure legend, which was missing, to help describe this figure.

Since this drug is important for helminths and there are studies looking at helminths and the microbiota, I think a sentence in the conclusions putting this into context would be helpful.

Excellent suggestion. We have added a sentence to the end of the paper.

Reviewer #2 (Remarks to the Author):

The authors report on the discovery of compounds, which block the toxic activity of *C. difficile* toxins A (TcdA) and B (TcdB). By screening of compounds, which are already used as drugs they identified niclosamide and 2 other related molecules as substances, which inhibited the cytopathic (cytotoxic) effects of TcdB. Niclosamide is an approved drug for worm diseases, which predominantly acts in the gut, because systemic up-take is minimal. They showed that niclosamide has no effect on the glucosylation of Rac protein or on toxin processing by inbuilt cysteine protease activity. Moreover, they report that niclosamide does not directly interact with TcdB or block toxin binding to cells. In contrast, they propose that niclosamide inhibits the acidification of endosomes, which is a prerequisite for toxin translocation into the cytosol of host cells. Not only cell up-take of TcdB and TcdA are blocked but also of CDT, which acts by a completely different toxin mechanism. They show that niclosamide or its more soluble preparation NEN protects mice towards *C. difficile*-induced pathogenesis. Moreover, they show that niclosamide does not destroy the normal microbiota as it is known (shown) for antibiotic therapy of *C. diff*-infection e.g. with vancomycin.

The finding of the potent action of niclosamide is of some interest. However, the precise mechanism of action of niclosamide has not been studied and clarified. Moreover, the paper contains several unclear statements. Specific comments

1. I miss time curves of the intoxication of cells. In some experiments it seems that niclosamide blocks completely the effect of TcdB (e.g., Fig. 1f and g). However, one need to have time courses of the intoxication process to see this in detail.

This is a good point. To address the time course of intoxication and inhibition in greater detail, we performed a kinetic TEER assay with a fixed dose of TcdB (5pM) and varying NEN concentrations and show the results in the new Supplementary Figure 2c. These data illustrate the kinetics of intoxication showing that after 5h there is complete loss of TEER and that NEN dose-dependently restores TEER at 7h.

2. Intoxication of cells was performed in serum free medium. This enhances generally the toxin activity. However, serum might completely block the effect of niclosamide. Therefore, also these conditions are important.

Indeed, this is very important and something that we considered carefully when designing these experiments. For the intoxication data presented in the manuscript, we chose to omit serum from the in vitro assay to better reflect conditions in the gut in vivo since there would not be expected to be any serum, or serum albumin for that matter, in the lumen of the gut. Interestingly, it has been speculated that protein binding that results once niclosamide and NEN enter the blood could contribute to the safety and tolerability in humans. As shown below, both niclosamide and FBS are less potent in the presence of FBS. These data have been added to the Supplementary Information with the creation of a new panel (d).

3. The picture in Fig. 1e is much too small and bars are missing.

Thank you for catching this. We have made Fig 1e larger, easier to see and have added scale-bars.

4. In general, the Fig. legends are much too short.

Yes, thank you for this comment. We have made substantial additions to the figure legends to include much more detail (highlighted in green in the attached revised manuscript).

5. Fig. 2f is strange. Why causes the addition of a low amount of TcdB such an increase in labeling by lysotracker. One would expect that also under control conditions without toxin low pH endosomal compartments exist, which should be labeled by lysotracker.

Excellent question. The apparent increase in labeling upon addition of TcdB, is a visual artifact that results from the decrease in the volume of cells intoxicated by TcdB. As a result of the decreased volume, the labeled lysosomes are brought closer together, giving the appearance of greater labeling. If one looks at the total fluorescence of the entire well, there is no change in total fluorescence (Tam et al., Chem Biol 2015 and see graph below). We have added significantly more detail to the figure legend to clarify the experiment.

6. The Fig. legend of Fig. 2 is incomplete. Only 2a and 2b are described. The legends for Fig. 2 c, d, e, f and g are missing.

We apologize for this oversight. We are not certain why it was not included in the initial submission, however, we have now included a full figure legend for Figure 2.

7. How was the necrosis measured in Fig. 2d?

The methods and newly revised figure legend now include a full description of how necrosis was measured.

8. The assay for the binding of TcdB is problematic (Fig. 2e), because the data may represent unspecific binding. This part is not to understand, because the legend of Fig. 2e is missing.

Once again, we apologize for not including this figure legend in the initial submission. The new figure legend explains how this assay was done and should clarify this concern. Also, we show in Fig 2e that only the control compound methyl cholate (described previously in PMID: 25619932) inhibited cell binding, whereas NEN did not. If there were any unspecific binding, one would not expect such a differential result.

9. The quality of Suppl. Fig. 4 is not satisfying.

The original picture quality was compromised during transfer into Illustrator. We have re-pasted the original TIFF images into a few file format, which are much clearer (see new S4 in Supplementary Information). In addition, we have added a new figure legend.

10. What is the reason for the poor systemic up-take of niclosamide, when it enters cells at low concentrations. Or is this effect on acidification of endosomal compartments only caused by endocytosis?

This is a great question. The ADME properties of niclosamide and NEN have been studied in depth by others, and though absorption is generally considered “poor”, it should be noted that there is in fact absorption, just not complete. Duhm et al showed that approximately one third of an oral NEN dose was absorbed after a 50 mg/kg oral dose – while two thirds were excreted as feces. What is likely is the colonocytes, which are the first layer of cells that Niclosamide and/or NEN encounter take up a large quantity and the amount of drug making it beyond decreases accordingly. To your second point, yes, there is very likely an additive effect of the cells being intoxicated and endocytosing both toxin and drug into cells also.

REVIEWERS' COMMENTS:

Reviewer #1 (Remarks to the Author):

The revised version of the manuscript entitled "Host-targeted niclosamide inhibits *C. difficile* virulence and prevents disease pathogenesis in mice without disrupting the gut microbiota" is much improved. The reviewers addressed most of my comments well and the added revisions to the manuscript improve the publication clarity and quality.

One of my main comments focused on the mouse model of CDI that was used, especially since the gut microbiota is not uniform in this model on Day 0. The authors agree that there are caveats to this model, but more importantly were still able to show a very significant result, where NEN treatment was able to rescue mice from death, or severe CDI. The comparisons between NEN and vancomycin are also important as vancomycin causes a significant perturbation to the microbiota compared to NEN.

I still feel this work is novel, significant, and will be well received by the wider scientific community.

Reviewer #2 (Remarks to the Author):

I have read the revised manuscript. The authors satisfyingly addressed all the questions from the referees.

Reviewers' comments: Responses are in RED BELOW

REVIEWERS' COMMENTS:

Reviewer #1 (Remarks to the Author):

The revised version of the manuscript entitled "Host-targeted niclosamide inhibits *C. difficile* virulence and prevents disease pathogenesis in mice without disrupting the gut microbiota" is much improved. The reviewers addressed most of my comments well and the added revisions to the manuscript improve the publication clarity and quality.

One of my main comments focused on the mouse model of CDI that was used, especially since the gut microbiota is not uniform in this model on Day 0. The authors agree that there are caveats to this model, but more importantly were still able to show a very significant result, where NEN treatment was able to rescue mice from death, or severe CDI. The comparisons between NEN and vancomycin are also important as vancomycin causes a significant perturbation to the microbiota compared to NEN.

I still feel this work is novel, significant, and will be well received by the wider scientific community.

Reviewer #2 (Remarks to the Author):

I have read the revised manuscript. The authors satisfyingly addressed all the questions from the referees.

→ We appreciate the comments and insightful comments/requests by reviewers 1 and 2.